# Solution-State Studies, X-ray Structure Determination and Luminescence Properties of an Ag(I) Heteroleptic Complex Containing 2,9-Bis(styryl)-1,10-phenanthroline Derivative and Triphenylphosphine

Dimitrios Glykos [1] , John C. Plakatouras [1,2] and Gerasimos Malandrinos [1,*]

[1] Laboratory of Inorganic Chemistry, Department of Chemistry, University of Ioannina, 451 10 Ioannina, Greece; d.glykos@uoi.gr (D.G.); iplakatu@uoi.gr (J.C.P.)

[2] Institute of Materials Science and Computing, University Research Center of Ioannina, 451 10 Ioannina, Greece

* Correspondence: gmalandr@uoi.gr

**Abstract:** A novel heteroleptic Ag(I) compound, formulated as [AgL(PPh$_3$)]BF$_4$ (1) (where L represents 2,9-bis((E)-4-methoxystyryl)-1,10-phenanthroline and PPh$_3$ stands for triphenylphosphine), was successfully synthesized and thoroughly characterized. The compound's stability in solution was confirmed through 1D and 2D nuclear magnetic resonance (NMR). The photo-irradiation of the complex in a CDCl$_3$ solution, utilizing a common portable UV lamp emitting at λ = 365 nm, led to the partial transformation of the E,E-geometric isomer to E,Z, ultimately yielding a 1:1.4 molar ratio of isomers. Its molecular structure was determined via X-ray crystallography, while molecular packing was assessed using Hirshfeld calculations. The most notable interactions (51%) within the cationic inner sphere involved H···H bonds. The photophysical characteristics of the complex and L were evaluated both in the solid state and in solution (dichloromethane). Compound **1** is a weak emitter, with photoluminescence quantum yields of 8.6% and 4.3% in solution and the solid state, respectively.

**Keywords:** Ag(I) coordination complex; X-ray; Hirshfeld analysis; photoluminescence; NMR studies; photo-isomerization

## 1. Introduction

Recent advancements in transition metal-based luminescent compounds have revolutionized the field of optoelectronics and sensing. Traditionally, heavy second- or third-row transition metals have been the cornerstone of luminescent complexes; however, their rarity and expense have spurred investigations into more cost-effective alternatives [1–4]. Transition metals with a d$^{10}$ electron configuration, particularly Cu(I), Ag(I), and Au(I), have emerged as promising candidates. These metals, known for their abundance and affordability, have demonstrated remarkable luminescent properties suitable for various applications. Notably, Cu(I) complexes have shown great potential in light-emitting devices and photovoltaics, signifying a shift toward sustainable and cost-effective technologies [5–10].

The research on luminescent single-metal Ag(I) complexes has intensified due to their potential utility in a vast range of applications, including sensing, photocatalysis, and light emission in various devices. Among these compounds, those featuring chelating aromatic diimine ligands (N^N) and phosphines (P or P^P) have been extensively investigated. Ag(I) complexes often exhibit phosphorescence originating from excited states centered on the ligands ($^3$LC) and, on occasion, display Thermally Activated Delayed Fluorescence (TADF) [11,12]. The electronic properties of diimine and phosphine ligands, along with the steric hindrance imposed by the insertion of bulky groups near the coordination sites, will influence both the nature of the excited states and the emitting properties ($\lambda_{em}$, photoluminescence quantum yield ($\Phi_{PL}$)) [13–20].

The literature has provided limited examples of luminescent Ag(I) trigonal complexes. Most of them are bimetallic with a diphosphine bridge [21–27], while mononuclear ones are even rarer [28–31]. Durini et al. recently reported several heteroleptic complexes of Ag(I) bearing the NˆN ligand 2-(1-(pyridin-2-yl)imidazo[1,5-a]pyridin-3-yl) phenol and various monodentate phosphine ligands. Notably, they successfully manipulated the emission maxima by altering the phosphine ligand. One of their compounds denoted as [Ag(NˆN)(PPh$_3$)][NO$_3$] exhibited intriguing photophysical characteristics, both in solid-state and solution environments (cyan to blue emitter) [32].

Moreover, 1,10-phenanthroline-based ligands have been the subject of extensive investigation over the past few decades. The distinctive structural and photophysical characteristics of the phenanthroline core have made ligands derived from it, along with their transition metal complexes, highly valuable scaffolds for a wide range of applications [33–35]. More specifically, styryl phenanthroline derivatives have been introduced, particularly for non-linear optics applications. By incorporating π-conjugated groups at the 2,9 positions of the central phenanthroline core, researchers have achieved high fluorescence efficiency. One effective strategy for achieving intriguing photophysical properties involves the use of molecules bearing D-π-A or D-π-A-π-D motifs, where D = electron donor, A = acceptor, and π = spacer. In this class of compounds, the fine-tuning of the optical properties can be easily achieved by the careful choice of electron-donor/-acceptor pairs and the π-conjugated spacer [36].

With the advantages of the above-mentioned derivatives and the promising photophysical characteristics of a simple Ag(I) trigonal complex [32] in mind, we believed it would be of interest to conduct the synthesis, characterization, and investigation of the photophysical attributes of a similar heteroleptic Ag(I) compound utilizing the diimine-type ligand L = 2,9-bis((E)-4-methoxystyryl)-1,10-phenanthroline and triphenylphosphine (PPh$_3$).

## 2. Results and Discussion

### 2.1. Characterization with NMR Spectroscopy

The $^1$H-NMR spectrum, as shown in Figure 1, was recorded using a CDCl$_3$ solution of the crystalline material, which had been used for X-ray studies. It exhibits a set of sharp and well-resolved signals, indicating the compound's integrity in the solution and the retention of C$_2$ symmetry on the NMR timescale. The sample was left in the NMR tube overnight, and a full set of 2D NMR experiments was scheduled for the next day.

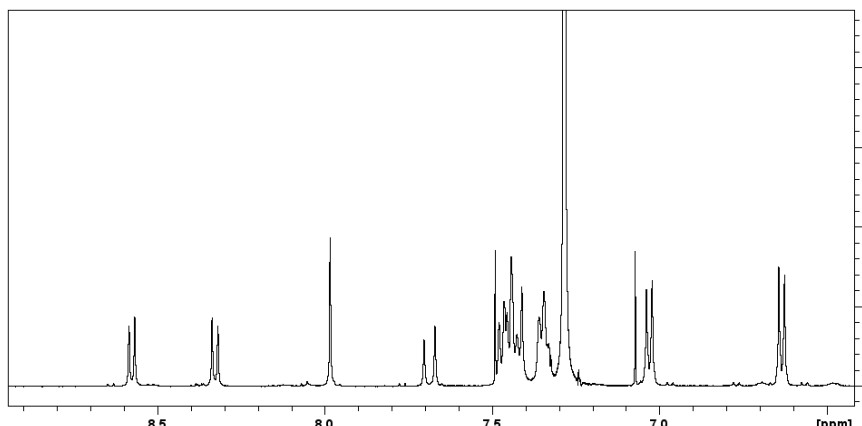

**Figure 1.** $^1$H-NMR spectrum (6.5–8.9 ppm) of 1 in CDCl$_3$ (500 MHz, 298 K) recorded using crystals of the compound.

Surprisingly, in the $^1$H-NMR spectrum of the "aged" sample (Figure S1a), some low-intensity peaks became apparent. Initially, we suspected a partial decomposition of the complex, potentially yielding L and/or the homoleptic species [AgL$_2$]$^+$. However, an

overlay of the spectra of this sample, the ligand, and $[AgL_2]^+$ (prepared in situ by mixing $AgBF_4$ with 2 eq of L) (Figure S2) ruled out this possibility.

Considering the photosensitivity of silver compounds, we decided to irradiate the $CDCl_3$ solution with a portable UV lamp emitting at $\lambda$ = 365 nm for a short period (10 min). The $^1$H-NMR spectrum in Figure S1b clearly shows a notable increase in the intensity of these extra peaks. An additional 30 min sample irradiation followed, resulting in the spectrum shown in Figure S1c.

An analysis of the 1–2D NMR data ($^1$H, $^{13}$C, COSY, NOESY, HSQC, HMBC) (Figures S1c, S3–S7) revealed that the E to Z photo-isomerization of one of the two C=C bonds took place, converting part of the complex from the (E,E) geometric isomer to the (E,Z) isomer. The E,E/E,Z molar ratio was calculated using the integral values of methoxy protons and found to be 1:1.43 (41:59%). This phenomenon is well-documented in organic styryl compounds [37–40], bis hydrazone [41], and metal complexes containing azo heteroarenes [42]. While there is one report in the literature for this ligand suggesting the presence of conformational isomers ($[CuL_2]^+$ species) [43], the detection of metallated 2,9-bis((E)-4-methoxystyryl)-1,10-phenanthroline geometric isomers due to photo-irradiation has never been reported.

The partial photo-isomerization of $[AgL(PPh_3)]^+$ (**1**) (E,E → E,Z) results in a $C_2$-symmetry breaking for the E,Z isomer and, consequently, (i) the presence of two 4-methoxystyryl environments and (ii) the chemical shift differentiation of all $^1$H and $^{13}$C atoms of the 1,10-phenanthroline core. Assigning the E,E isomer was straightforward, but the E,Z case was more challenging. The most important correlations (2D NMR) that aided in accomplishing this task are provided in Figures S8–S11, with the $^1$H-NMR spectrum, including a complete assignment of all peaks, presented in Figure 2 (the atom labels used for the assignments are shown in Scheme 1).

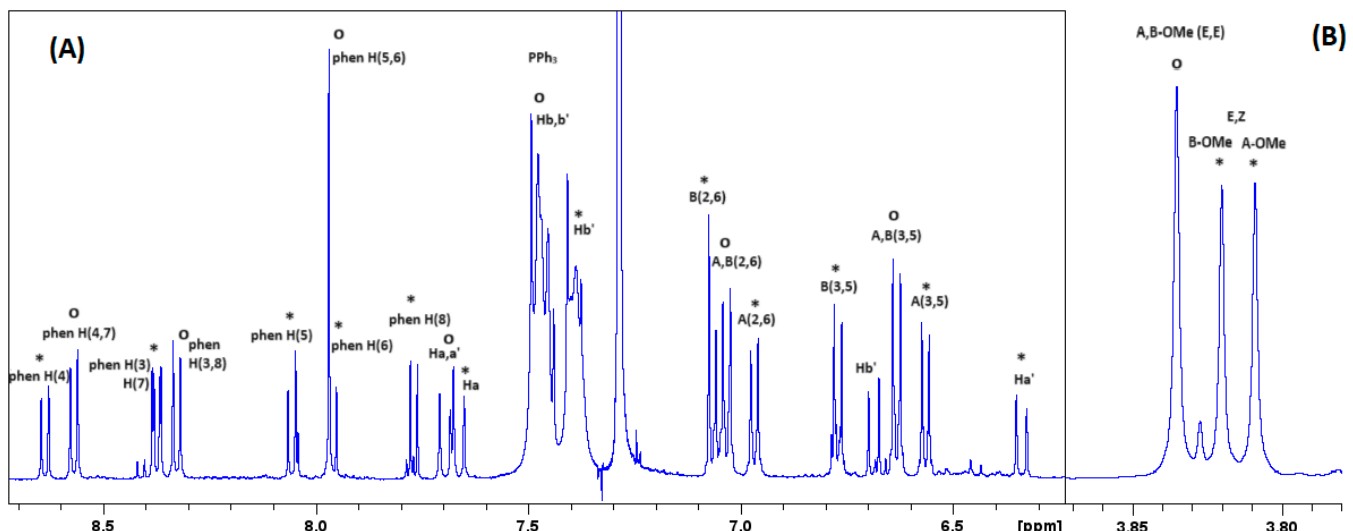

**Figure 2.** $^1$H-NMR spectrum of a $CDCl_3$ solution of **1** (E,E and E,Z isomers) including peak assignments. Signals belonging to E,E are marked by a circle while E,Z's are marked by an asterisk. (**A**): aromatic region (**B**): methoxy region. See Scheme 1 for atom labeling.

The $^1$H and $^{13}$C chemical shifts ($\delta$, ppm) and chemical shift differences ($\Delta\delta = \delta_{complex} - \delta_L$) derived from the above analysis are listed in Table 1 and Table S1, respectively. These data, particularly the $\Delta\delta$ values, clearly illustrate that the complexation of both ligands with Ag(I) causes a significant perturbation of the chemical shift for all protons belonging to either the phen core (downfield) or the 4-methoxy rings A and B (upfield). In the latter case, the observed shifts should be attributed to the anisotropic shielding effect created by the aromatic $PPh_3$ rings. When comparing the $\Delta\delta$ values of 1-(E,E) and 1-(E,Z), the most significant difference is observed for phen H(8) (0.4 vs. −0.16 ppm, see Table 1), which

was expected due to the conformational change (E → Z) of the double bond linking the phen-4-methoxyphenyl B ring.

**Scheme 1.** Atom numbering in L (E,E) and (E,Z) isomers for NMR spectroscopic assignments.

**Table 1.** $^1$H-NMR data ($\delta$,ppm) for L/PPh$_3$, and complex (E,E and E, Z isomers).

| H Atoms | L (E,E)/PPh$_3$ | Complex (E, E) | $\Delta\delta$ (E,E) * | H Atoms | Complex (E,Z) | $\Delta\delta$ (E,Z) * |
|---|---|---|---|---|---|---|
| H(3) | 7.93 | 8.33 | 0.4 | H(3) | 8.37 | 0.44 |
| H(4) | 8.22 | 8.57 | 0.35 | H(4) | 8.64 | 0.42 |
| H(5) | 7.74 | 7.97 | 0.23 | H(5) | 8.06 | 0.32 |
| H(6) | 7.74 | 7.97 | 0.23 | H(6) | 7.96 | 0.22 |
| H(7) | 8.22 | 8.57 | 0.35 | H(7) | 8.37 | 0.15 |
| H(8) | 7.93 | 8.33 | 0.4 | H(8) | 7.77 | −0.16 |
| H(a,a') (E) | 7.77 | 7.70 | −0.07 | H(a) (E) | 7.67 | −0.1 |
| H(b,b') (E) | 7.62 | 7.43 | −0.19 | H(b) (E) | 7.39 | −0.23 |
| | | | | H(a') (Z) | 6.34 | −1.43 |
| | | | | H(b') (Z) | 6.69 | −0.93 |
| A, H(2,6) | 7.68 | 7.04 | −0.64 | A H(2,6) | 6.97 | −0.71 |
| A, H(3,5) | 6.99 | 6.63 | −0.36 | A H(3,5) | 6.56 | −0.43 |
| B, H(2,6) | 7.68 | 7.04 | −0.64 | B H(2,6) | 7.07 | −0.61 |
| B, H(3,5) | 6.99 | 6.63 | −0.36 | B H(3,5) | 6.77 | −0.22 |
| A, (-OMe) | 3.9 | 3.84 | −0.06 | A H(-OMe) | 3.81 | −0.09 |
| B, (-OMe) | 3.9 | 3.84 | −0.06 | B H(-OMe) | 3.82 | −0.08 |
| o,p-PPh$_3$ | 7.32–7.38 | 7.47, 7.48 | - | o,p-PPh$_3$ | 7.47, 7.48 | - |
| m-PPh$_3$ | | 7.39 | - | m-PPh$_3$ | 7.39 | - |

* $\Delta\delta = \delta_{complex}$ (E,E or E,Z) − $\delta_{ligand}$.

Additional information concerning the Ag(I) coordination sphere and geometry was obtained through $^{31}$P-NMR spectroscopy. The spectrum of **1**, recorded in CDCl$_3$, is shown in Figure 3.

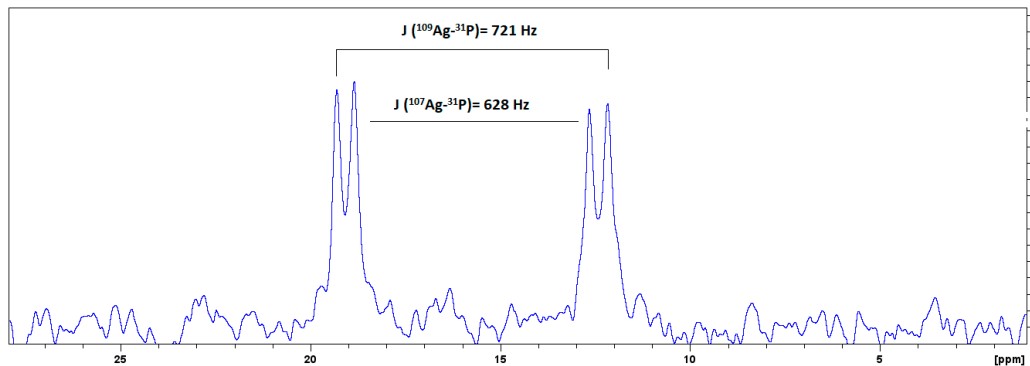

**Figure 3.** $^{31}$P{$^1$H}-NMR spectrum of **1** in CDCl$_3$ (101.25 MHz, 298 K).

A well-resolved doublet of doublets centered at 15.75 ppm appears, from which the $J(^{109}$Ag-$^{31}$P) and $J(^{107}$Ag-$^{31}$P) couplings can be calculated (721, 628 Hz, respectively). The values of J couplings imply that Ag(I) adopts a trigonal geometry employing the diimine ligand and the phosphine [32]. The irradiation process causes no significant change in the $^{31}$P chemical shift (16.44 ppm). The signal appears now broad and unstructured, possibly due to the slightly different $^{31}$P chemical environments of the two isomers (Figure S12).

Lastly, we decided to investigate if the photo-isomerization reaction could proceed further. An additional 2 h irradiation produced no significant spectral changes (Figure S13). Only a 2% increment of the E,Z percentage in the mixture was observed (new E,E/E,Z molar ratio: 1:1.54). Thus, we assume that under the reaction conditions used, the equilibrium state has been reached.

## 2.2. Optical Properties in Solution

The UV-vis and emission spectra of complex 1 in CH$_2$Cl$_2$ are both depicted in Figure 4, with detailed photophysical data available in Table 2. It exhibits absorption bands at around 240 nm and within the 330–380 nm wavelength range, with no noticeable absorption observed beyond 450 nm. The former is attributed to ligand-centered (LC) $\pi \rightarrow \pi^*$ transitions, while the bands observed in the 330–380 nm spectral range are expected to have a combined character of L-to-L Charge Transfer (LLCT) and M-to-L Charge Transfer (MLCT). This behavior is commonly observed in complexes of the type M[(N^N)(P)] (where M represents Ag(I) or Cu(I)) [13,16,44,45].

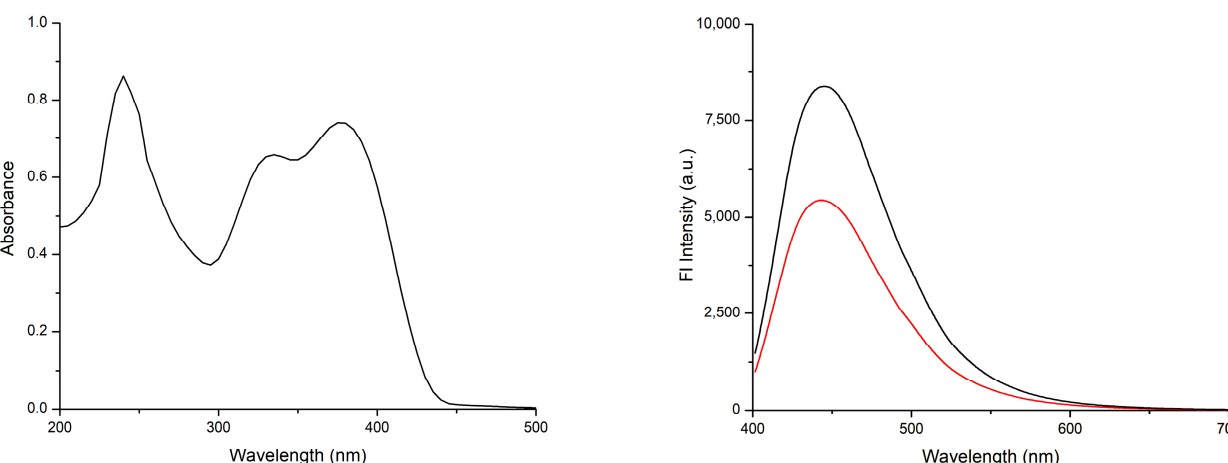

**Figure 4.** The UV-vis spectrum of complex **1** (on the left) and the emission spectra of complex **1** (black trace) and ligand L (red trace) in dichloromethane ($10^{-5}$ M) (on the right).

**Table 2.** Selected photophysical data for compound **1** and the ligand in dichloromethane ($10^{-5}$ M) and in the solid state.

| | Absorption | | Emission (Solution) | | | Emission (Solid) | | |
|---|---|---|---|---|---|---|---|---|
| | $\lambda_{abs}$ (nm) | $\varepsilon$ ($M^{-1}cm^{-1}$) | $\lambda_{exc}$ (nm) | $\lambda_{em}$ (nm) | $\Phi_{PL}$ (%) | $\lambda_{exc}$ (nm) | $\lambda_{em}$ (nm) | $\Phi_{PL}$ (%) |
| Ligand | 365 | 55,000 | 365 | 433 | 6 | 365 | 570 | 3.8 |
| **1** | 375 | 4008 | 375 | 446 | 8.6 | 375 | 500 | 4.3 |

The relative photoluminescent quantum yield ($\Phi_{PL}$) calculated for **1** in solution is 8.6%. Notably, this value does not significantly differ from the quantum yield obtained for the ligand L, which is 6%.

### 2.3. Absorption Spectrum (DRS)

Figure 5 depicts the diffuse reflectance spectrum (DRS) of **1**. The broad spectral pattern ranging from 240 to 480 nm is commonly found in such complexes. Electronic transitions associated with ligand-centered (LC) $\pi \rightarrow \pi^*$ and $n \rightarrow \pi^*$ transitions are expected within the 240–350 nm range, while lower-energy bands (above 350 nm) can be ascribed to LLCT and MLCT transitions [46–48].

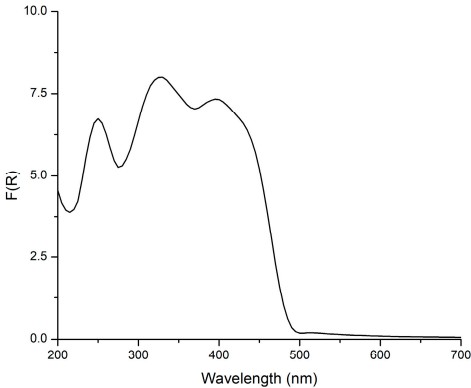

**Figure 5.** The Kubelka–Munk spectrum of complex **1**.

### 2.4. Luminescent Behavior in the Solid State

The emission spectrum of compound **1** in the solid state is displayed in Figure 6. It shows a hypsochromic shift in the emission maxima compared to the ligand L, with a shift of $\Delta\lambda$ = 70 nm. The absolute photoluminescent quantum yield ($\Phi_{PL}$) for **1** is relatively low at 4.3%, closely resembling that of L (3.8%). However, the noted hypsochromic shift mentioned earlier suggests that Ag(I)-L interaction may have an impact on the nature of the excited state.

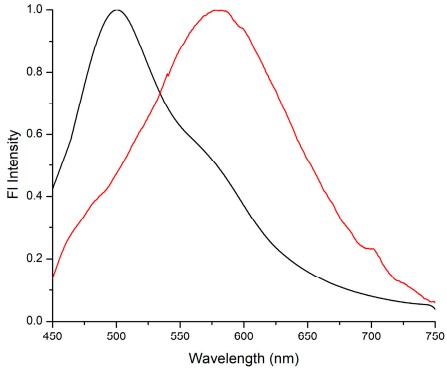

**Figure 6.** Normalized emission spectra of complex **1** ($\lambda_{exc}$ = 375 nm, black trace) and ligand L ($\lambda_{exc}$ = 365 nm, red trace) in the solid state.

## 2.5. IR Spectroscopy

In the ATR-IR spectrum of compound **1**, as depicted in Figure S14, the characteristic bands assigned to C=N, B-F and C-P stretching vibrations (1580, 1050 and 517 cm$^{-1}$ respectively) are clearly observable, indicating the presence of L, PPh$_3$ and BF$_4^-$ [17].

## 2.6. Description of the Structure

Compound **1** crystallizes in the triclinic space group P-1. The asymmetric unit consists of a positively charged cation [AgL(PPh$_3$)]$^+$ and the corresponding counter-anion, BF$_4^-$. The visual representation of this cation's structure can be found in Figure 7. Selected bond distances (in Ångstroms) and angles (in degrees) for the coordination sphere of Ag(I) in the cation are presented in Table 3.

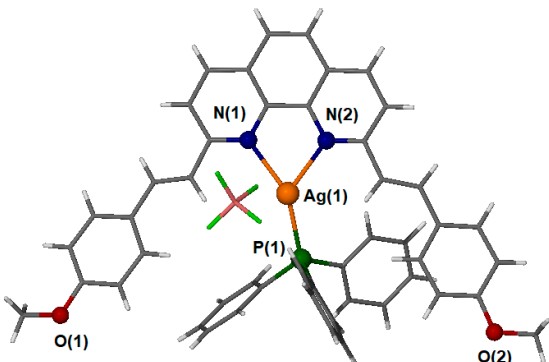

**Figure 7.** A ''ball and stick'' presentation of compound **1** with a partial labeling scheme.

**Table 3.** Selected structural characteristics of [AgL(PPh$_3$)]BF$_4$.

| Bond Distances | (Å) | Bond Angles | (°) |
|---|---|---|---|
| Ag(1)-N(1) | 2.287(2) | N(1)-Ag(1)-N(2) | 73.26(8) |
| Ag(1)-N(2) | 2.336(2) | N(1)-Ag(1)-P(1) | 151.34(6) |
| Ag(1)-P(1) | 2.3743(7) | N(2)-Ag(1)-P(1) | 134.03(6) |

Compound **1** belongs to a limited group of compounds that have undergone structural analysis and are described by the formula [Ag(N-N chelate)(unidentate phosphine)](counter anion) [30,31,49]. Notably, it stands out as the first known example containing the ligand 2,9-bis((E)-4-methoxystyryl)-1,10-phenanthroline. In the cationic part of the structure, Ag(1) is positioned at the center of a distorted trigonal planar geometry. This geometry is formed by two nitrogen atoms, N(1) and N(2), belonging to a chelating ligand, and one phosphorus atom, P(1), which is part of a coordinated PPh$_3$ molecule. The deviations from the ideal trigonal planar arrangement are evident in the relatively wide N(1)-Ag(1)-P(1) angle of 151.34(6)°, primarily due to the substantial steric hindrance imposed by the phenyl groups. Additionally, the acute N(1)-Ag(1)-N(2) angle of 73.26(8)° arises from the small bite angle of chelation. The bond distances between Ag and P, as well as Ag and N, align with values previously reported in the literature [30,49]. The weaker interaction of the ligand with silver(I), compared to that of the literature complex [Ag(phen)(PPh$_3$)]CF$_3$SO$_3$, was confirmed by inspecting the difference in Ag-N(2) bond length (mean values: Ag-N(2)$_{phen}$, 2.280; Ag-N(2)$_L$, 2.336 Å).

The compound displays an offset face-to-face π-stacking motif between phenanthroline rings. The corresponding distances from centroid-to-centroid and centroid-to-plane are 3.61 and 3.43 Å, respectively, with a ring offset of 1.15 Å [50,51].

A packing diagram illustrating the π-stacking and phenyl embraces between two independent PPh$_3$ molecules is shown in Figure 8.

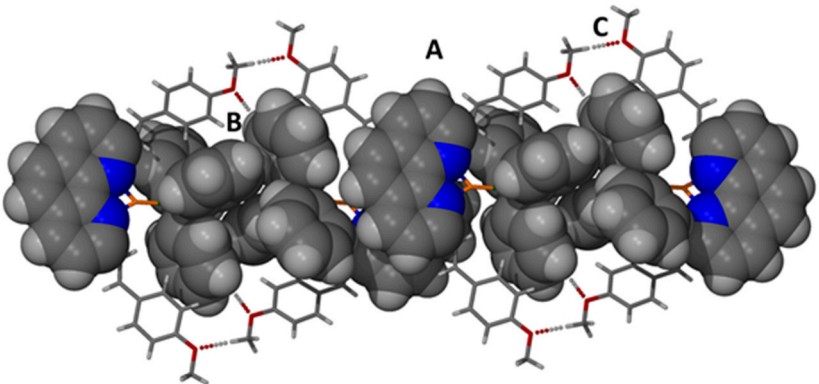

**Figure 8.** A part of the packing in the crystal structure of the prepared compound showing the dominant weak interactions present: stacking between the phen moieties (**A**); phenyl embraces between triphenylphosphines' phenyl rings (**B**); non-conventional C–H ⋯ O hydrogen bonds (**C**). Symmetry operations to generate equivalent atoms: (**A**), −x + 1, −y + 1, −z; (**B**), x, y, z − 1; (**C**): −x + 1, y + 1, −z + 1. A packing diagram of [3,2,0] direction of the unit cell.

### 2.7. Hirshfeld Surface Analysis

A convenient and accurate method for discerning the various interactions among atoms in a crystal structure is Hirshfeld Surface Analysis [52]. This method was employed to dissect the intermolecular interactions within the crystal structure of the [AgL(PPh₃)]BF₄ complex. The generated Hirshfeld maps are depicted in Figure 9. Interestingly, even though the tetrafluoroborate counter anion was not directly involved in the Ag(I) coordination sphere, it plays a significant role in intermolecular contact.

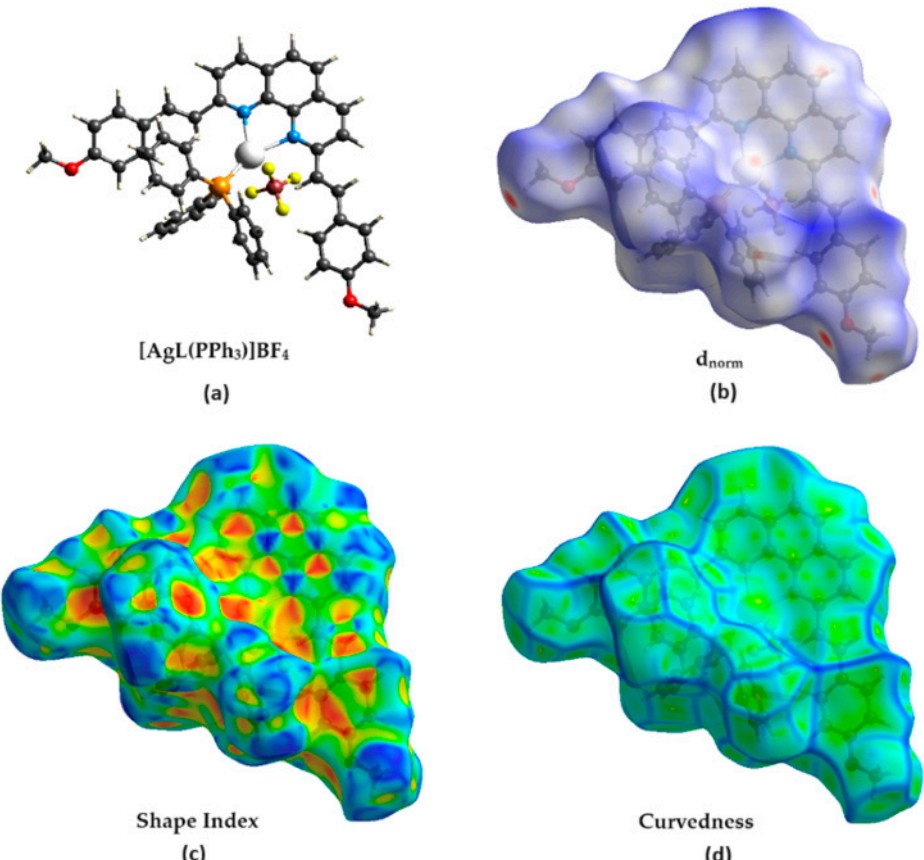

**Figure 9.** Hirshfeld surfaces for complex **1** (**a**): mapped over d$_{norm}$ (**b**); shape index (**c**); and curvedness (**d**).

The 2D fingerprint plots for 1, shown in Figure S15, demonstrate that H···H contacts contribute the largest portion (50.1%) to the total Hirshfeld surface at the range of de + di ≈ 2.2 Å. The second-largest contribution belongs to C···H/H···C contacts with the tips at de + di ≈ 3.2 Å. The contribution of O···H/H···O contacts appears as a diffuse shape with de + di ≈ 2.6 Å. In the shape index map, the presence of red and blue triangles indicates aromatic stacking interactions.

Furthermore, the flat green areas in the curvedness map further support the existence of π-stacking interactions. The percentage of C···C interactions was calculated to be 6.4%. The other weak contacts (Ag···H/H···Ag) and (Ag···C/C···Ag) are identified as low-density scattered points in 2D fingerprint plots. Lastly, the crystal holds $BF_4^-$ through strong intermolecular interactions, including non-conventional C-H···F hydrogen bonds.

## 3. Materials and Methods

### 3.1. Materials

All solvents were of analytical purity and used as received. $AgBF_4$, $PPh_3$ and 4-anisaldehyde were purchased from Sigma-Aldrich (Burlington, MA, USA), while neocuproine was purchased from TCI Chemicals (Tokyo, Japan). All reactions were carried out under an inert atmosphere unless otherwise stated.

### 3.2. Methods

Unless otherwise stated, all spectroscopic work was conducted using crystals of the isolated compound (E,E-isomer). The crystalline material was dissolved in the appropriate solvent, and the spectra were recorded immediately. To check if the excitation radiation used in the emission spectrum could promote the partial transformation of the E,E isomer to E,Z, a [1]H-NMR spectrum was acquired. The spectrum is depicted in Figure S16 and shows that the compound remains intact (please compare with Figure 1).

A high-resolution–electrospray ionization–mass spectrum (HR-ESI-MS) was acquired using a Thermo Scientific LTQ Orbitrap XL™ system. One-dimensional ([1]H, [13]C) and two-dimensional (COSY, NOESY, HMBC, HSQC) nuclear magnetic resonance (NMR) experiments were conducted on a Bruker Avance spectrometer (Bruker Biospin GmbH, Ettlingen, Germany) with proton and carbon frequencies of 500.13 MHz and 125.03 MHz, respectively. Additionally, $^{31}P\{^1H\}$ NMR spectra (101.25 MHz) were recorded at room temperature on a Bruker Avance spectrometer (Bruker Biospin GmbH, Ettlingen, Germany) with a proton frequency of 250.13 MHz. The chemical shifts for [1]H and [13]C were referenced to the residual solvent peak ($CDCl_3$), while [31]P shifts were referenced to an external standard (85% $H_3PO_4$). NMR data were processed using Topspin 4.07 (Bruker Biospin GmbH, Ettlingen, Germany). An Agilent Cary 630 ATR-IR (Harrick Scientific Products, Inc., New York, NY, USA) spectrometer was utilized to record the infrared spectrum, and an Agilent Cary 60 UV–vis spectrophotometer (Harrick Scientific Products, Inc., New York, NY, USA) was employed to register both the solution and solid-state (DRS) spectra of the compound. Emission studies were conducted using the infrastructure and procedures outlined in previous works [44,53].

### 3.3. Crystal Structure Determination

For the collection of the diffraction data, an appropriately sized yellow prism-shaped crystal (dimensions: $0.45 \times 0.22 \times 0.18$ mm$^3$) was mounted on the goniometer of a Bruker D8 Quest Eco diffractometer (Ettlinger, Germany) equipped with a Photon II detector and a TRIUMPH (curved graphite) monochromator, utilizing Mo Kα radiation (wavelength λ = 0.71073 Å) [54]. A wide-frame method was used to integrate the gathered frames, which included φ and ω scans. The multi-scan approach (SADABS (Version 2.03)) [55] was applied to rectify the data (correction for absorption effects). The structure was solved by direct methods, and the ShelXle interface (Version Qt5-64-1.0.1506) enabled full-matrix least-squares methodology to be used on F$^2$ (SHELXL 2018/3) [56,57]. Non-H atoms were subjected to anisotropic treatment, and organic H atoms were positioned in the ideal

positions based on calculations and then refined as riding on their respective carbon atoms. PLATON (Version 2023.1) was utilized for geometric computations [58], while X-Seed (Version 4.10) was employed to produce molecular visualizations [59].

Crystal data for $C_{48}H_{39}AgN_2O_2PBF_4$ (M = 901.46 g/mol) are summarized as follows: triclinic, space group P-1 (no. 2), a = 10.2795(3) Å, b = 11.8100(3) Å, c = 18.7503(4) Å, $\alpha$ = 98.070(2), $\beta$ = 103.124(2), $\gamma$ = 106.865(2), V = 2068.90(10) Å$^3$, Z = 2, T = 296(2) K, $\mu$(MoK$\alpha$) = 0.576 mm$^{-1}$, Dcalc = 1.447 g/cm$^3$, 68,781 reflections measured (2.50 < $\theta$ < 27.54), 7274 unique (Rint = 0.0578), which were used in all calculations. The final R1 was 0.0324 (I > 2$\sigma$(I)), and wR2 was 0.0790 (all data) (Table S2).

CCDC2303493 contains the supplementary crystallographic data for this paper. These data can be obtained free of charge via https://www.ccdc.cam.ac.uk/structures/? (accessed on 25 October 2023).

### 3.4. Synthesis

3.4.1. Synthesis of Ligand L

The ligand L 2,9-bis((E)-4-methoxystyryl)-1,10-phenanthroline was prepared according to the literature procedure [36]. $^1$H and $^{13}$C data are given in Table S3.

3.4.2. Synthesis of [AgL(PPh$_3$)]BF$_4$ Complex

A 25 mL round-bottom flask containing 10 mL of a 5:2 *v/v* CH$_2$Cl$_2$/MeOH mixture was charged with 26.2 mg of PPh$_3$ (0.1 mmol) and 19.67 mg of AgBF$_4$ (0.1 mmol). The clear solution was stirred for 2 h under argon at room temperature. Subsequently, 44.45 mg of L (0.1 mmol) was added, and the resulting yellow solution was stirred for an additional 2 h. The crude product, isolated by evaporating to dryness, was washed with diethyl ether, collected, and dried under vacuum, resulting in a yield of 79%. Single crystals suitable for X-ray analysis were obtained by the vapor diffusion of diethyl ether into a CH$_2$Cl$_2$ solution of the compound. The material, verified through X-ray crystallography, is complex 1 (E,E-isomer). Although we did not analyze the solution from which the crystals were obtained, we believe that UV-light absorption by the glass jar used to crystallize the compound (under ambient light) and the higher thermodynamic stability of the E,E-isomer might explain its selective isolation. $C_{48}H_{39}AgN_2O_2PBF_4$ (E,E isomer): $^1$H NMR (500 MHz, CDCl$_3$) (ppm): 8.57 (d, *J* = 8.6 Hz, 2H); 8.33 (d, *J* = 8.6 Hz, 2H); 7.97 (s, 2H); 7.70 (d, *J* = 16.3 Hz, 4H); 7.43 (d, *J* = 16.3 Hz, 4H); 7.04 (d, *J* = 8.7 Hz, 4H); 6.63 (d, *J* = 8.7 Hz, 4H); 3.84 (s, 3H); PPh$_3$ 7.39–7.47 (m). $^{13}$C NMR (125 MHz, CDCl$_3$) (ppm) Table S1: 160.7; 156.4; 142.6; 139.4; 138.5; 129; 127.9; 126.5; 126.4; 121.7; 114.3; 55.4 (L), 129.5; 130.4; 131.5; 133.8 (PPh$_3$).

$C_{48}H_{39}AgN_2O_2PBF_4$ (E,Z isomer): $^1$H NMR (500 MHz, CDCl$_3$) (ppm): 8.64 (d, *J* = 8.6 Hz); 8.37 (d, *J* = 8.6 Hz); 8.37 (d, *J* = 8.4 Hz); 8.06 (d, *J* = 8.7 Hz); 7.96 (d, *J* = 8.7 Hz); 7.77 (d, *J* = 8.4 Hz); 7.67 (d, *J* = 16.2 Hz); 7.39 (d, *J* = 16.2 Hz); 7.07 (d, *J* = 8.7 Hz); 6.97 (d, *J* = 8.7 Hz); 6.77 (d, *J* = 8.7 Hz); 6.69 (d, *J* = 12 Hz); 6.56 (d, *J* = 8.7 Hz); 6.34 (d, *J* = 12 Hz); 3.82 (s); 3.81 (s) PPh$_3$ 7.39–7.47 (m). $^{13}$C NMR (125 MHz, CDCl$_3$) (ppm) Table S1: 160.7; 160.2; 157; 156.7; 142.6; 142.3; 139.6; 138.9; 138.7; 136.6; 130.6; 129.1; 128.5; 128.4; 127.8; 127.7; 127.3; 127.2; 126.4; 126.3; 125.5; 122.1; 114.3; 114.2; 55.4; 55.3. (L), 129.5; 130.4; 131.5; 133.8 (PPh$_3$).

$^{31}$P NMR (101.25 MHz), CDCl$_3$) (ppm): 15.75 (dd, *J*($^{109}$Ag-$^{31}$P) = 721 Hz, *J*($^{107}$Ag-$^{31}$P) = 628 Hz).

HR ESI-MS: *m/z* = 815.1801 for [AgL(PPh$_3$)]$^+$, additional fragments present: *m/z* = 515.0885 for [AgL]$^+$, and *m/z* = 997.2723 for [AgL$_2$]$^+$ (Figures S17–S20).

## 4. Conclusions

In summary, we have successfully synthesized and characterized a novel Ag(I) heteroleptic complex (1) containing the diimine-type ligand 2,9-bis((E)-4-methoxystyryl)-1,10-phenanthroline and triphenylphosphine. X-ray crystallography has revealed that Ag(I) adopts a distorted trigonal planar geometry, formed by the two chelating diimine nitrogen atoms and one phosphorus atom. This structural arrangement is also retained in solution, as indicated by the observable $^{107}$/$^{109}$Ag-$^{31}$P coupling constants in the $^{31}$P{$^1$H}-NMR spectrum.

Photo-irradiation at 365 nm of the compound in $CDCl_3$ induces photo-isomerization, resulting in a mixture of E,E and E,Z geometrical isomers, which have been thoroughly characterized in solution. Notably, 1 is a weak emitter both in the solid state and in solution, exhibiting photoluminescent quantum yields of 4.3% and 8.6%, respectively.

We intend to continue our studies by synthesizing more similar complexes and investigating their photo-reactivity in greater detail.

**Supplementary Materials:** The following supporting information can be downloaded at: https://www.mdpi.com/article/10.3390/inorganics11120467/s1, Figure S1: $^1$H-NMR spectra (aromatic region) of 1 in $CDCl_3$ solutions (500 MHz, 298 K): (a) an "aged" sample, (b) irradiated at $\lambda = 365$ nm for 10 min, (c) irradiated for an additional 30 min; Figure S2: $^1$H-NMR spectra (aromatic region) of $CDCl_3$ solutions of: (a) 1 irradiated at $\lambda = 365$ nm for 40 min, (b) L, (c) $[AgL_2]^+$; Figure S3: $^{13}$C{$^1$H}-NMR spectrum of a $CDCl_3$ solution of 1 irradiated at $\lambda = 365$ nm for 40 min (125 MHz, 298 K); Figure S4: $^1$H-$^1$H-COSY NMR spectrum (500 MHz, 298 K) of a $CDCl_3$ solution of 1 irradiated at $\lambda = 365$ nm for 40 min; Figure S5: $^1$H-$^1$H-NOESY NMR spectrum (500 MHz, 298 K) of a $CDCl_3$ solution of 1 irradiated at $\lambda = 365$ nm for 40 min; Figure S6: $^1$H-$^{13}$C-HSQC NMR spectrum (298 K) of a CDCl3 solution of 1 irradiated at $\lambda = 365$ nm for 40 min; Figure S7: $^1$H$-^{13}$C-HMBC NMR spectrum (500 MHz, 298 K) of a $CDCl_3$ solution of 1 irradiated at $\lambda = 365$ nm for 40 min; Figure S8: Overlay of $^1$H-$^1$H-COSY (blue) and $^1$H-$^1$H-NOESY (red-magenta) NMR spectra of a $CDCl_3$ solution of 1 irradiated at $\lambda = 365$ nm for 40 min. The most informative correlations toward identification of the phen core $^1$H's are circled; Figure S9: Overlay of $^1$H-$^1$H-COSY (blue) and $^1$H-$^1$H-NOESY (red-magenta) NMR spectra of a $CDCl_3$ solution of 1 irradiated at $\lambda = 365$ nm for 40 min, showing the NOE correlations from ethylenic protons; Figure S10: Part of $^1$H-$^1$H-NOESY spectrum of 1 irradiated at $\lambda = 365$ nm for 40 min, showing the methoxy-A,B(3,5) protons correlations for both isomers. See Scheme 1 for atom labeling; Figure S11: Part of $^1$H-$^{13}$C-HSQC spectrum of 1 irradiated at $\lambda = 365$ nm for 40 min, showing correlations of H(3)/C(3), H(7)/C(7) for E,Z isomer and H(3,8)/C(3,8) for E,E isomer. See Scheme 1 for atom labeling; Figure S12: $^{31}$P{$^1$H}-NMR spectra of $CDCl_3$ solutions of **1** (101.25 MHz, 298 K): (a) using crystals of the compound; (b) irradiated at $\lambda = 365$ nm for 40 min; Figure S13: $^1$H-NMR spectrum (aromatic region) of a $CDCl_3$ solution of 1 irradiated at $\lambda = 365$ nm for 2 h (298 K); Figure S14: The ATR-IR spectrum of **1**; Figure S15: 2D fingerprint plots of $[AgL(PPh_3)]BF_4$; Figure S16: $^1$H-NMR spectrum (aromatic region) of a $CDCl_3$ solution of 1 acquired immediately after recording the emission spectrum (298 K);Figure S17: HR-ESI-MS spectrum of compound **1** in $CHCl_3$; Figure S18: HR-ESI-MS spectrum of the fragment $[AgL]^+$ (top) and theoretical spectrum; Figure S19: HR-ESI-MS spectrum of the fragment $[AgL(PPh_3)]+$ (top) and theoretical spectrum; Figure S20: HR-ESI-MS spectrum of the fragment $[AgL_2]^+$ (top) and theoretical spectrum; Table S1: $^{13}$C-NMR data ($\delta$,ppm) for $L/PPh_3$, and complex (E,E and E, Z isomers); Table S2: Crystal data and structure refinement for C48 H39 Ag B F4 N2 O2 P at 296(2) K.; Table S3: $^1$H and $^{13}$C NMR data ($\delta$,ppm) for the ligand; File S1: crystal structure of the compound (*.cif file); File S2 (checkcif, pdf file).

**Author Contributions:** Conceptualization: G.M.; supervision: G.M.; formal analysis: D.G. and J.C.P.; investigation: D.G.; X-ray crystallography: J.C.P. and D.G.; writing—original draft: D.G.; writing—review and editing: G.M. and J.C.P. All authors have read and agreed to the published version of the manuscript.

**Funding:** This research received no external funding.

**Data Availability Statement:** The information provided in this research is accessible in both the article and its Supplementary Materials.

**Acknowledgments:** The authors would like to thank the Network of Research Supporting Laboratories at the University of Ioannina for providing access to the use of NMR, HR ESI-MS and X-ray diffraction facilities.

**Conflicts of Interest:** The authors declare no conflict of interest.

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
