# Peer review of "Solution-State Studies, X-ray Structure Determination and Luminescence Properties of an Ag(I) Heteroleptic Complex Containing 2,9-Bis(styryl)-1,10-phenanthroline Derivative and Triphenylphosphine"

_inorganics, doi:10.3390/inorganics11120467_

Round 1
Reviewer 1 Report
Comments and Suggestions for Authors
This paper considers the novel heteroleptic complex [AgL(PPh3)]BF4 where L repre- 11 sents 2,9-bis((E)-4-methoxystyryl)-1,10-phenanthroline.
The crystal structure of the complex has been obtained. The structure has been well refined and the cif file is complete. The description of the structure is well done.
My only concern is with Figure 8 which purports to show the π-stacking and phenyl embraces between two independent PPh3 molecules. However the figure is totally confusing and no such interaction is apparent. There must be a better view with fewer molecules and less overlap.
Analysis of the 1-2D NMR data revealed that E to Z photo-isomerization of one of the two C=C bonds can take place, converting part of the complex from the symmetric (E,E) isomer to the (E,Z) isomer.
Details of other physical methods are obtained and thoroughly discussed.
This is a good paper, with sound science and well written.
Author Response
We would like to express our gratitude to Reviewer 1 for taking the time to study and provide comments on our manuscript. We have incorporated a new Figure 8, along with its caption (highlighted in yellow color), to better illustrate the dominant weak interactions present in the crystal structure.
Reviewer 2 Report
Comments and Suggestions for Authors
This manuscript depicted about structural and spectral features of a triangle silver complex coordinated by three donor atoms, diimine with branches containing a C=C double bond and monophosphine. Especially, the story of isomerization is much more attractive, I think. All the more, some interpretations in the description of the isomerization include ambiguity which should be explained exactly, I also think.
1) The manuscript said that E,E/E,Z molar ratio was found to be 1:1.43 (41:59 %). Can I understand that the isomerization reaction is equilibrium between Z,Z and Z,E? Can not Z,E isomer be formed further in additional irradiation? It should be depicted how large the ratio of the isomers reached finally.
2) In “3.4 synthesis” section, NMR data was depicted for E,E and E,Z isomers. Is it sure that these data were obtained by measurements of the mixture of two isomers? Can I understand that the separation was not successful and only E, E isomer was isolated by recrystallization and characterized in the structural features? If so, why do you think only E,E isomer was obtained without isomerization during recrystallization? I desire that any answer to it is contained in this manuscript.
3) On the measurement of the emission spectra, is it sure that the isomerization from Z,Z to Z,E did not really occur on irradiation of excitation light? In NMR, the experiment of isomerization was carried out by irradiation at 365 nm. Though this wavelength was different from the excitation wavelength in emission spectra( 375 nm), it was similar to each other. So I wonder if the isomerization has occurred on the measurement and the emission spectra of mixed two isomers had been observed. It is necessary to prove that only Z,Z isomers are present in the measured solution with some kinds of methods, I think.
Author Response
We would like to express our gratitude to Reviewer 2 for the time and effort spent evaluating the manuscript and providing valuable comments and suggestions. We respond as follows:
Comment1: The manuscript said that E,E/E,Z molar ratio was found to be 1:1.43 (41:59 %). Can I understand that the isomerization reaction is equilibrium between Z,Z and Z,E? Can not Z,E isomer be formed further in additional irradiation? It should be depicted how large the ratio of the isomers reached finally.
Answer: We have performed a new NMR experiment in which an additional 2h irradiation was applied. There were no significant spectral changes (Fig. S13). Only a 2% increment of the E,Z percentage in the mixture was observed (new E,E/E,Z molar ratio: 1:1.54). Thus, we assume that under the reaction conditions used, the equilibrium state has been reached. This result has been added to the NMR section (highlighted in blue color).
Comment2: In “3.4 synthesis” section, NMR data was depicted for E,E and E,Z isomers. Is it sure that these data were obtained by measurements of the mixture of two isomers? Can I understand that the separation was not successful and only E, E isomer was isolated by recrystallization and characterized in the structural features? If so, why do you think only E,E isomer was obtained without isomerization during recrystallization? I desire that any answer to it is contained in this manuscript.
Answer: Yes, NMR data have been obtained by analyzing the mixture of the two isomers in CDCl3. As described in the NMR section, this mixture results after irradiation of a CDCl3 solution containing the crystalline material. Our goal was to identify the origin of the extra 1H peaks emerging after irradiation, and in the course of analysis, we found that they belong to the E,Z-geometric isomer. Therefore, no attempt was made to isolate the mixture contents. The crystallization process involves the slow diffusion of Et2O vapor into a CH2Cl2 solution of the compound under ambient light in a glass jar (glass absorbs UV-light). Although we did not analyze the solution from which the crystals were obtained, we believe that UV-light absorption by the glass jar used for the crystallization process (under ambient light) and the higher thermodynamic stability of the E,E-isomer might explain its selective isolation. This possible explanation has been added to the manuscript (section 3.4, highlighted in blue color).
Comment3: On the measurement of the emission spectra, is it sure that the isomerization from Z,Z to Z,E did not really occur on irradiation of excitation light? In NMR, the experiment of isomerization was carried out by irradiation at 365 nm. Though this wavelength was different from the excitation wavelength in emission spectra( 375 nm), it was similar to each other. So I wonder if the isomerization has occurred on the measurement and the emission spectra of mixed two isomers had been observed. It is necessary to prove that only Z,Z isomers are present in the measured solution with some kinds of methods, I think.
Answer: All spectroscopic work in solution (UV-Vis, Emission Spectroscopy, HR ESI-MS) was conducted using crystals of the isolated compound (E,E-isomer). The crystalline material was dissolved in the appropriate solvent, and the spectra were recorded immediately. To check if the excitation radiation used in the emission spectrum could promote partial transformation of the E,E isomer to E,Z, a 1H-NMR spectrum was acquired. The spectrum is depicted in Fig. S16 and shows that the compound remains intact (please compare with Fig. 1). We have added this paragraph to the Materials and Methods section (highlighted in blue color).